# Hypergraph Multi-modal Large Language Model: Exploiting EEG and Eye-tracking Modalities to Evaluate Heterogeneous Responses for Video Understanding

Minghui Wu[*†]
Chenxu Zhao[*]
Anyang Su[*]
wuminghui@mininglamp.com
zhaochenxu@mininglamp.com
suanyang@mininglamp.com
Mininglamp Technology
Beijing, China

Donglin Di
Tianyu Fu
Da An
donglin.ddl@gmail.com
futianyu0514@126.com
anda93456@gmail.com
Shanghai Artificial Intelligence Lab
Shanghai, China

Min He
hemin@mininglamp.com
Mininglamp Technology
Beijing, China

Ya Gao
gaoya@stu.pku.edu.cn
Peking University
Beijing, China

Meng Ma
mameng@pku.edu.cn
Peking University
Beijing, China

Kun Yan[‡]
Ping Wang[‡]
kyan2018@pku.edu.cn
pwang@pku.edu.cn
Peking University
Beijing, China

## Abstract

Understanding of video creativity and content often varies among individuals, with differences in focal points and cognitive levels across different ages, experiences, and genders. There is currently a lack of research in this area, and most existing benchmarks suffer from several drawbacks: 1) a limited number of modalities and answers with restrictive length; 2) the content and scenarios within the videos are excessively monotonous, transmitting allegories and emotions that are overly simplistic. To bridge the gap to real-world applications, we introduce a large-scale Video **S**ubjective **M**ulti-modal **E**valuation dataset, namely Video-SME. Specifically, we collected real changes in Electroencephalographic (EEG) and eye-tracking regions from different demographics while they viewed identical video content. Utilizing this multi-modal dataset, we developed tasks and protocols to analyze and evaluate the extent of cognitive understanding of video content among different users. Along with the dataset, we designed a **H**ypergraph **M**ulti-modal **L**arge **L**anguage **M**odel (HMLLM) to explore the associations among different demographics, video elements, EEG and eye-tracking indicators. HMLLM could bridge semantic gaps across rich modalities and integrate information beyond different modalities to perform logical reasoning. Extensive experimental evaluations on Video-SME and other additional video-based generative performance benchmarks demonstrate the effectiveness of our method. The code and dataset are available at this url.

## CCS Concepts

• **Computing methodologies** → **Video summarization**.

## Keywords

Video Understanding, Electroencephalographic, Multi-modal Large Language Model, Hypergraph Learning

**ACM Reference Format:**
Minghui Wu, Chenxu Zhao, Anyang Su, Donglin Di, Tianyu Fu, Da An, Min He, Ya Gao, Meng Ma, Kun Yan, and Ping Wang. 2024. Hypergraph Multi-modal Large Language Model: Exploiting EEG and Eye-tracking Modalities to Evaluate Heterogeneous Responses for Video Understanding. In *Proceedings of the 32nd ACM International Conference on Multimedia (MM '24), October 28–November 1, 2024, Melbourne, VIC, Australia.* ACM, New York, NY, USA, 14 pages. https://doi.org/10.1145/3664647.3680810

[*]These authors contributed equally to this research.
[†]Also with Peking University.
[‡]Corresponding author.

## 1 Introduction

With the advancement of Large Language Models (LLMs) [72] and Multi-modal Large Language Models [11, 32, 45, 46], the field of video understanding has entered a new era. The advanced logical reasoning abilities of multi-modal LLMs facilitate a thorough analysis of explicit elements within videos. Moreover, these models can deduce the underlying implicit content of these explicit factors, leveraging the knowledge and experience acquired by LLMs. Existing benchmarks for video content question-and-answering, such as [31, 53, 75, 75, 79], provide a rich set of instruction labels. Alternatively, they exhibit several deficiencies as illustrated in Table 1: 1) the video content itself is overly simplistic, often only involving objective, explicit factors, which does not support the exploration of deeper levels of video creativity and implicit factors.

**Table 1: Comparison of existing VideoQ&A datasets with ours (OE: open-ended, MC: multiple-choice, AP: Audience Profiles).**

| Datasets | Video source | Q&A generation | Q&A tasks | Modality | Videos | Q&A pairs | AvgAnsLen | MedScene |
|---|---|---|---|---|---|---|---|---|
| MSVD-QA [75] | MSVD | Auto | OE | Video | 1,970 | 50,505 | 1.0 | 2 |
| MSRVTT-QA [75] | MSRVTT | Auto | OE | Video | 10,000 | **243,680** | 1.0 | 3 |
| TGIF-QA [31] | TGIF | Auto&Human | OE & MC | Frame/Video | **56,720** | 103,919 | 1.5 | 1 |
| ActivityNet-QA [79] | ActivityNet | Human | OE | Video | 5,800 | 58,000 | 1.3 | 7 |
| Video-ChatGPT [53] | ActivityNet | Auto&Human | OE | Video | 200 | 2,994 | 51.0 | 6 |
| Video-SME-QA (ours) | Custom | Auto&Human | MC & OE | **Video/EEG/EMR/AP** | 498 | 178,547 | **99.6** | **11** |

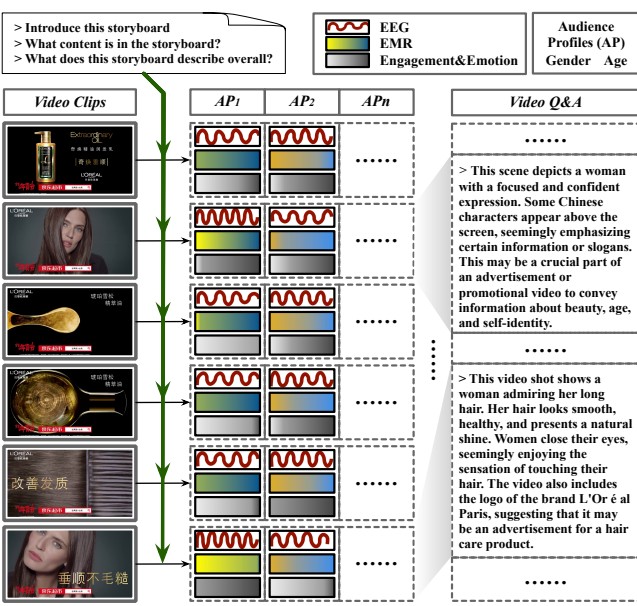

**Figure 1: Our proposed Video Subjective Multi-modal Evaluation (Video-SME) dataset for Subjective Response Indicators (SRI). Real-time signals captured by electroencephalographic (EEG) and eye-tracking devices reveal that Audience Profiles (AP) of varying genders and ages exhibit distinct engagements, emotions, and eye motion ratios (EMR) when exposed to various scenes and elements within the same advertisement video.**

We utilize the **MedScene** metric to evaluate this issue, where **MedScene** denotes the median number of scene across all videos in the dataset. A higher number of scenes indicates greater complexity in video content; 2) the number of modalities included in these datasets are limited, generally confined to videos and frames; 3) the instruction labels concerning the length of answers are restricted to certain predetermined options, failing to assess the divergent and analytical abilities of LLMs. We utilize the **AvgAnsLen** to evaluate this issue, where **AvgAnsLen** represents the average text length of the answer portion across all Q&A pairs in the dataset. To address the issues mentioned above, we have prepared an extensive collection of content-rich advertisement videos, accompanied by a more comprehensive set of modality labels.

In the burgeoning field of cognitive neuroscience, the exploration of how individuals perceive and interpret video content has opened new avenues for understanding the intricate interplay between brain activity and media interaction [67]. Recent advancements in multi-modal data analysis have underscored the importance of leveraging diverse physiological signals to gain insights into the cognitive and emotional states of viewers [37]. Among these, Electroencephalographic (EEG) signals with their high temporal resolution, provide a direct measure of brain activity [59], capturing the nuanced and dynamic changes in cognitive states as individuals engage with video content. These signals embody the electrical manifestations of the brain's complex neural dynamics, offering insights into the emotional and cognitive processes underpinning video content interpretation [58].

Inspired by the aforementioned context, we have utilized EEG and eye-tracking apparatus to collect and record the EEG and eye movement responses of individuals across various ages, genders, and professions while watching the same advertisement video. We aggregated this information into modality labels, introducing a novel, large-scale benchmark: Video Subjective Multi-modal Evaluation dataset, namely **Video-SME**. As illustrated in Figure 1, our proposed dataset captures the subjective reactions of individuals watching videos through EEG and eye-tracking devices, fills the gaps in the video understanding domain regarding the assessment of video appeal and implicit factors. How to effectively leveraging these multi-modal labels to uncover the latent associations among the modalities becomes the cornerstone for addressing deeper challenges in video understanding.

Graph-based methodologies exhibit superiority in exploring the associations among features, particularly hypergraphs, extending beyond traditional graph theory, offer a powerful framework for representing complex relationships in data [6]. In the context of video content analysis, hypergraphs can encapsulate the intricate associations among video elements, EEG signals, and eye-tracking data, allowing for the modeling of higher-order interactions that are not capturable through simple pairwise connections.

Utilizing the multi-modal information of the Video-SME dataset, coupled with the superiority of constructing associative features through hypergraph, we proposed a Hypergraph Multi-modal Large Language Model (**HMLLM**), integrating information from disparate modalities to perform logical reasoning and semantic analysis. By leveraging the rich information encoded in video content, along with EEG and eye-tracking data, HMLLM can bridge semantic gaps across modalities, offering a comprehensive understanding of the cognitive processes involved in video content interpretation.

The main contributions can be summarized as follows:

1. Introduction of a novel large-scale benchmark dataset: the Video Subjective Multi-modal Evaluation (Video-SME) dataset, a large-scale benchmark that captures real-time EEG and eye-tracking data from a diverse demographic while they watch advertisement

videos. This dataset fills a significant gap in the field of video understanding by providing rich modality information and a comprehensive set of question-and-answer (Q&A) pairs that allow for the assessment of video creativity and implicit factors.

2. Development of the Hypergraph Multi-modal Large Language Model (HMLLM): we have developed a novel HMLLM that leverages the complex relationships among video elements, EEG signals, and eye-tracking data encapsulated in hypergraphs.

3. Extensive experimental evaluations demonstrating our method's effectiveness: through rigorous experimental evaluations conducted on the Video-SME dataset and additional video Q&A datasets, we have demonstrated the effectiveness of our HMLLM.

## 2 Preliminaries

### 2.1 Video Understanding

Video understanding aims to create algorithms that allow machines to interpret videos with the same expertise as humans. Meanwhile, video emotion recognition [44, 57, 84] emphasizes the interplay between the emotions conveyed by the video and the viewer responses, collectively forming a critical component of video understanding. Most existing works focus on modeling objective and tangible visual properties of videos [16], particularly in action recognition [3, 7, 10, 17, 19, 20, 54, 61, 69, 71] and temporal action localization/detection [18, 49, 86]. However, the need for content recommendation systems has spurred research into subjective and intangible aspects (e.g. the appeal and memorability of content [14]), where various semantically rich information are considered [5, 13, 29, 55, 85].

Compared with the above work, we present a new large-scale dataset filled with content-rich advertisement videos. This dataset includes a wider range of labels that cover both tangible and intangible aspects of content. Leveraging this dataset, we introduce an advanced hypergraph multi-modal large language model. This model is designed to simultaneously process various modalities, enabling it to conduct logical reasoning and perform in-depth semantic analysis of video content.

### 2.2 EEG-Based Emotion Recognition

Electroencephalography (EEG) signals provide detailed insights into brain activity related to emotions, offering spatial information on specific brain regions involved [8]. The Arousal-Valence model [60] is a key framework for classifying emotions along two dimensions. Xiaolin et al [63] explored various features to enhance the emotion recognition model. However, there's a shift towards deep learning due to the limitations of machine learning. The dynamical graph convolutional neural network (DGCNN) [62] was proposed to learn discriminative EEG features and interrelationships among EEG channels. Some works have moved towards multi-modal learning for robust results in EEG signal recognition tasks, such as integrating physiological signals in the multi-modal framework to enhance emotion recognition accuracy [74], and employing proper windowing and channel selection to avoid relying on the full length of EEG and EOG signals for classification [9]. Furthermore, advancements in neuromorphic computing led to the use of Spiking Neural Networks (SNN) [52] for classifying spatiotemporal EEG data with lower computational requirements [35].

### 2.3 Multi-modal Large Language Models

Multi-modal Large Language Models (MLLMs), primarily serving as vision-language models, transform images or videos into texts. These models are mainly divided into two categories: traditional large-scale pretraining [39, 40, 68] and instruction tuning using pre-trained LLMs [50, 78, 87]. The first category comprises models that blend a visual encoder with a language model, either developed from scratch or based on pre-existing models, possibly including a trainable module to bridge the two modalities. Utilizing auto-regressive loss for text generation, these models are training on extensive image-text datasets, including image-text pairs [27, 39, 40, 68] and image-text sequence instances [2]. The second category, drawing inspiration from instruction-tuning techniques used in MLLMs [1, 56], incorporates instruction-following data to enhance MLLMs' zero- and few-shot learning abilities [15, 50, 78, 87]. A notable example is LLaVA [50], which employs a simple projection matrix to link a pre-trained visual encoder with an LLM, focusing initially on pre-training for feature alignment before comprehensive end-to-end fine-tuning. Some other works extend to video understanding by connecting video encoders to MLLMs [41, 47, 77, 81]. In addition to models that focus on combining images or videos with text, there are projects that incorporate even more types of data, like speech, audio, and sensor information [25, 64, 73, 80].

### 2.4 Hypergraph Learning

A hypergraph includes vertices and hyperedges, where hyperedges can connect multiple vertices. This structure is more adaptable and effective for representing complex relationships in data than traditional graphs [24]. Methods for creating hypergraphs fall into two groups: explicit and implicit. Explicit methods directly use the data structure to form hyperedges, like connecting vertices with shared attributes [28, 34]. Implicit methods, however, infer hyperedges from data without clear high-order links, utilizing approaches based on distance [22] or representations [33, 48, 51, 70]. Unlike static structures, some methods allow for hypergraph structure optimization, adjusting it during the learning phase. This involves adaptively changing weights on hyperedges [23] or sub-hypergraphs [83] to improve learning outcomes. Recent advancements have introduced deep hypergraph representation learning, a new approach that mainly divides into spectral [21, 76] and spatial [4, 26] categories based on how hypergraph convolution operator is defined.

## 3 Video-SME Dataset

In this section, we present the Video Subjective Multi-modal Evaluation (**Video-SME**) dataset. The Video-SME dataset not only focuses on the Objectivity Task typically found in traditional video Q&A datasets but also meticulously collects Subjective Response Indicators (**SRI**) to enhance the richness. It encompasses a wide array of advertisement videos across different industries. To capture a diverse set of responses, we enlisted participants from various cities throughout Mainland China. These participants are equipped with EEG devices, enabling us to monitor their brainwave activities and eye motion ratios (EMR) in real-time while watching the advertisements. The collected data is subsequently analyzed to establish a benchmark for the classification of brainwave and EMR responses, which is elaborated in Sections 3.1 and 3.2.

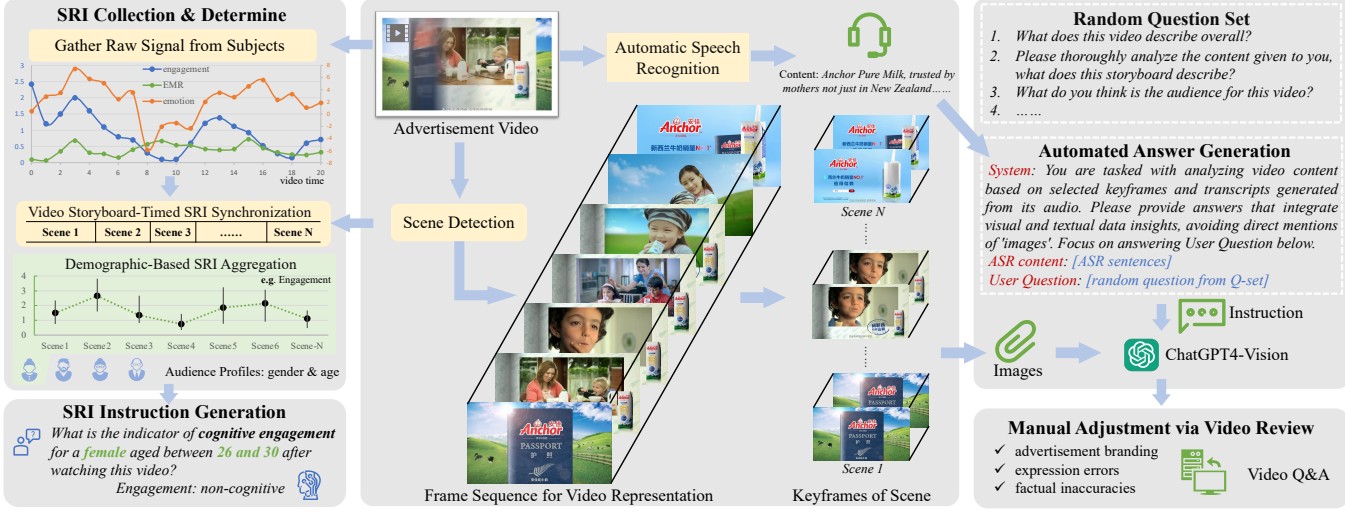

Figure 2: Generation pipeline of Video-SME dataset. The left side of this figure illustrates the process of SRI data collection, computation, and amalgamation. This involves acquiring raw signals from subjects, processing signals by video scenes, and pooling data from subjects with similar demographic profiles to obtain aggregated Subjective Response Indicators (SRI) and instruction for language models. The middle section depicts the video preprocessing with Frame Sequence for Video Representation (FSVR) by scene detection and Automatic Speech Recognition (ASR) for videos. On the right side, we present our proposed semi-automated video Q&A generation process, which leverages both video storyboarding from FSVR and dialogue text from ASR. This integration enriches video content comprehension, facilitating both Subjectivity and Objectivity Tasks.

Additionally, the Video-SME dataset includes an extensive video Q&A section to provide objective insights into the ads, facilitating model training and subjective index assessment. The task definition and protocol of our dataset are outlined in Section 3.3 and 3.4.

## 3.1 Frame Sequence for Video Representation

The Video-SME dataset features Chinese advertising videos from diverse fields such as food and beverages, household items, consumer electronics, cultural tourism, software, and automobiles. It comprises 498 curated landscape videos sourced from online platforms and TV commercial ads, each running for 15-30 seconds.

In this study, we introduce the Frame Sequence for Video Representation (**FSVR**) strategy to preprocess advertisement videos, as depicted in the middle part of Figure 2. We enhance the video scene sensitivity by integrating the AdaptiveDetector[1] for FSVR with specific parameters: adaptive_threshold = 2, min_scene_len = 10, window_width = 2. In the case of advertisement videos with frequent scene changes, the scene detection algorithm captures more information compared to average frame capture methods. Moreover, it is invaluable in minimizing redundant frames in videos primarily composed of static scenes.

By employing FSVR, we are able to deconstruct the temporal sequence of advertisement video frames, achieving capabilities including modality signal alignment, video content understanding, and semi–automated Q&A instruction generation.

## 3.2 Subjectivity: SRI Collection & Classification

We developed a sophisticated system for collecting subjective indicators. Each participant watches a series of advertisement videos using the device described in the appendix. During this process, we synchronously gather EEG and eye-tracking data, along with anonymized demographic details. Our study includes over 4,600 participants, ensuring a wide demographic representation. The participant base spans white-collar workers, civil servants, students, and freelancers across various age groups and income brackets.

The raw EEG signals are characterized by parameters such as $\alpha_1, \alpha_2 \ldots \beta_2, \beta_3$ [36, 38], which is detailed in the appendix. Given the unique demands of advertisement video analysis, we pinpointed two pivotal EEG metrics: engagement and emotion, as delineated by Equation 1 and Equation 2, respectively.

$$EN_t = (\beta_2 + \beta_3) / (\alpha_3 + \alpha_2 + \beta_2 + \beta_3), \qquad (1)$$

$$EM_t = (\alpha_3 - \alpha_2) / (\alpha_3 + \alpha_2) \times 100, \qquad (2)$$

where $EN_t$ and $EM_t$ represent the engagement and emotion of the individual user at the sampling moment, respectively. Furthermore, we tracked eye movement data, defining the Eye Movement Ratio ($EMR_t$) as the proportion of time the participant's gaze fixates on the display relative to the total video duration.

The SRI Collection & Determine workflow, depicted on the left of Figure 2, captures sub-second high-frequency raw signals data. To align with video content's scene-based evolution, Video Storyboard-Timed SRI Synchronization was adopted, producing time-averaged and participant-specific SRIs. Demographic characteristics then grouped these SRIs into units of 5-20 same-gender participants with

---

[1]https://www.scenedetect.com/

a maximum age difference of 5 years, such as {female, <20}, {male, 26-30}, and {female, 46-50}, as Demographic-Based SRI Aggregation in Equation 3.

$$\bar{X} = \frac{1}{P \cdot N} \sum_{i=1}^{P} \sum_{j=1}^{N} X_{p_i, t_j}, \forall p_i \in [AP], \forall t_j \in [t_1, t_2], \quad (3)$$

where $X_{p_i, t_j}$ denotes the original SRI such as $EN_t$, $EM_t$, and $EMR_t$. Each indicator associated with discrete values for participant $p_i$ at specific timestamps $t_j$, where $t_j$ signifies the effective sampling moment instances within the video storyboard timeframe from FSVR in Section 3.1.

For quantitative analysis, we meticulously examined data distribution across various Audience Profile segments. Engagement was categorized into two groups using the Leuven Engagement Scale (LES) and its distribution. Emotion and EMR indicators, which followed normal distributions, were divided into three equal categories. For detailed data distribution, refer to the appendix. The SRI Instruction Generation protocol is detailed in Table 2.

### 3.3 Objectivity: Semi-automated Generation

In addition to subjective indicators from Audience Profiles, we developed a semi-automated annotation pipeline for ChatGPT4-Vision (GPT4V) to obtain Objective Video Q&A, depicted in Figure 2. Although GPT4V cannot process videos, it supports multiple consecutive key-frames simultaneously. Based on FSVR in Video Preprocessing, we extracted middle frames from each shot as keyframes that effectively represent the entire video. During each invocation of GPT4V to automatically generate answers, questions are selected randomly from the Random Question Set to enhance the diversity of Q&A sessions, along with providing ASR text and FSVR key-frames. Lastly, annotators were carefully selected to manually refine objective Q&A instruction from Automated Answer Generation, addressing issues like advertisement branding, expression errors, and factual inaccuracies.

### 3.4 Data Overview, Tasks and Protocols

Based on the processing presented in Sections 3.2 and 3.3, Video-SME is categorized into subjectivity and objectivity tasks. The subjectivity task examines the SRI, whereas the objectivity task is dedicated to the qualitative analysis of video content and audience perception. As shown in Table 2, we present the tasks, protocols, and instructions associated with the Video-SME dataset.

Task 1, entitled **Subjectivity**, is formulated as a classification task, aimed at examining the influence of video content and user characteristics on the SRI. We develop two experimental protocols to guide this investigation. The first protocol (**P1**) is designed to assess the SRI ability of a broad audience, involving the analysis of average responses across different videos. This approach is relatively straightforward. The second protocol (**P2**) introduces a layer of complexity by focusing on the SRI discernment of particular user demographics. This necessitates a comprehensive examination of how response patterns fluctuate among diverse user cohorts.

Task 2, designated as **Objectivity**, mirrors the video Q&A tasks prevalent in prior datasets, as described in Section 3.3. Building on the method outlined in [53], this study conducts a supervised analysis of the answers generated, assessing their accuracy and

**Table 2: Task and Protocol of Video-SME Dataset. In Task1, Protocol1 (P1) targets a broad audience. Protocol2 (P2), based on P1, contains SRI to Audience Profiles.**

| Task Name | 1. Subjectivity | | 2. Objectivity |
|---|---|---|---|
| **Eva. Form** | Multi-classification | | Text generation |
| **Train Video** | 426 | | 426 |
| **Test Video** | 72 | | 72 |
| **Train Q&A** | 145,107 | | 5762 |
| **Test Protocol** | P1 | P2 | – |
| **Test Q&A** | 2,640 | 26,724 | 954 |

allocating scores. This approach is designed to objectively ascertain the narrative coherence of the advertisement content and its efficacy in captivating the target audiences.

## 4 Method

This section elaborates on the Hypergraph Multi-modal Large Language Model (HMLLM), an approach designed to intelligently process video clips and textual prompts for generating contextually relevant text, including Subjective Response Indicators (SRI). Central to our methodology are several key components as depicted in Figure 3: Visual Encoder, Query Former (Q-Former), SALM Projector, SRI-Aware Language Model (SALM), and SAL-HL Module. All components mentioned above synergistically orchestrated across two primary phases: SALM Warm-Up and SAL-HL Fine-Tuning, as depicted in our model architecture (refer to Figure 3). The pseudocode in the appendix illustrates the detailed training process.

### 4.1 SALM Warm Up

We begin by detailing the initial stage. The approach ingests brief video clips and corresponding textual prompts, extracting key frames from the videos using a predefined, static extraction strategy, which can be either random or uniformly distributed. These key frames are represented as $F = \{f_0, f_1, \ldots, f_N\}$, with $N$ signifying the number of extracted frames. These key frames are then pre-processed to form the initial data matrix, denoted by $\mathbf{X}_0 \in \mathbb{R}^{B \times C \times N \times h \times w}$, where $B$, $C$, $N$, $h$, and $w$ correspond to the batch size, color channels (RGB), the number of keyframes, and the resized dimensions of the frames, respectively. The initial data matrix $\mathbf{X}_0$ is fed into a pre-trained visual encoder to yield initial visual representations, expressed as $\mathbf{F}_v \in \mathbb{R}^{B \times N \times F_L \times F_C}$, with $F_L$ and $F_C$ representing the length and channels of features, respectively.

During the first training phase, the "Hypergraph Learning Gate (HL-Gate)" remains inactive while the Q-Former and SALM are warmed up. The visual features $\mathbf{F}_v$ are then input into the frozen Q-Former as the Key ($\mathbf{K} \in \mathbb{R}^{B \times (N \times F_L) \times F_C}$) and Value ($\mathbf{V} \in \mathbb{R}^{B \times (N \times F_L)}$) for the attention mechanism. The Query in the Q-Former is initialized as either a random or null set, represented by $\mathbf{Q} \in \mathbb{R}^{B \times (Q \times C_q)}$, where $Q \times C_q$ are the predefined hyper-parameters for the length and channels of the query. Subsequently, we introduce an "SALM Projector", a multi-layer perceptron that follows the Q-Former, capable of reshaping the data and introducing additional learning parameters into the model. The output of projector is denoted

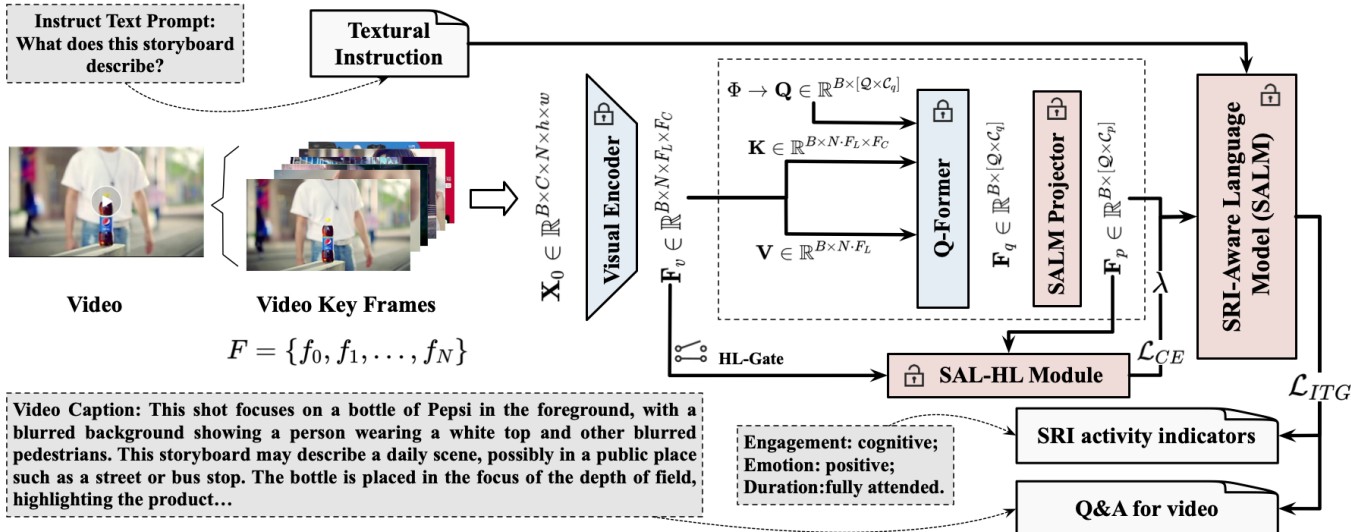

**Figure 3: Overview of the HMLLM. The architecture comprises a suite of pre-trained models, including a "Visual Encoder", "Q-Former", and the "SRI-Aware Language Model (SALM)", which are initially frozen and subsequently fine-tuned through strategic training procedures. More importantly, our model incorporates a designed "SRI-Aware Language Hypergraph Learning (SAL-HL)" module that is trained de novo via a combined loss function. During inference, the HMLLM generates SRI and Q&A responses tailored to the video content, thereby providing a deeper level of engagement and comprehension.**

as $\mathbf{F}_p \in \mathbb{R}^{B \times (Q \times C_p)}$, with $C_p$ being another predefined hyper-parameter. The SRI-Aware Language Model (SALM) is then engaged, taking the output of the SALM Projector ($\mathbf{F}_p$) and the corresponding textual instructions as inputs during the initial warm-up training stage. The SALM is trained using the Image-grounded Text Generation (ITG) loss function [40] ($\mathcal{L}_{ITG}$), which instructs the Q-Former to generate text conditioned on the input images. The goal of the ITG loss is to minimize the difference between the generated caption $\widetilde{\mathcal{Y}}_{qa} \leftarrow \text{SALM}(\mathbf{F}_p, T)$ and the ground-truth caption $\mathbf{Y}_{gt}$. This is typically achieved using a cross-entropy loss computed over the words or tokens in the caption. The ITG loss function can be mathematically represented as:

$$\mathcal{L}_{ITG} = -\sum \left( \log \mathbb{P}(\mathbf{Y}_{gt_i} | \mathbf{Y}_{gt_1}, \cdots, \mathbf{Y}_{gt_{i-1}}, \mathbf{F}_v) \right) \quad (4)$$

where $\mathbb{P}(\mathbf{Y}_{gt_i} | \mathbf{Y}_{gt_1}, \cdots, \mathbf{Y}_{gt_{i-1}}, \mathbf{F}_v)$ denotes the probability of generating the $i$-th word in the caption given the previous words and the visual features extracted from the image. The summation encompasses all words or tokens in the ground-truth caption.

In our approach, we integrate specific strategies from BLIP2 [40] to address the limitation of Q-Former architecture on direct interactions between the image encoder and text tokens. Following the aforementioned training procedure, the SALM Projector and SALM are adequately warmed up, preparing them for subsequent fine-tuning optimization.

### 4.2 SAL-HL Fine-tune

In the subsequent fine-tuning phase, the hypergraph learning gate (HL-Gate) is activated, and the hypergraph learning module (SAL-HL) undergoes training in tandem with the fine-tuning of the SRI-Aware Language Model (SALM). As delineated in Figure 3, the

SAL-HL module receives the initial visual features ($\mathbf{F}_v$) and the representations of the projected frames ($\mathbf{F}_p$) produced by the warmed SALM Projector as inputs.

The SAL-HL module initiates the process by merging these two feature sets (i.e., $\mathbf{F}_p$, $\mathbf{F}_v$) and then pooling them to generate frame-level representations ($\mathbf{F}_{frame\_level}$). This process is formulated as:

$$\mathbf{F}_{frame\_level} = \text{Pool} \left( \text{Feature\_Mixer} \left( \mathbf{F}_p \mathbf{F}_v \right) \right). \quad (5)$$

The *Feature_Mixer* denotes the mixing operation between two feature matrices, which can be implemented as a multi-layer perceptron (MLP). Each frame, denoted as $f_i$ for $i \in [0, N]$, is considered a vertex ($\mathcal{V}$) within the hypergraph structure ($\mathcal{G}$), which facilitates the establishment of high-order relationships among the frames. The construction of the hypergraph entails the application of a clustering algorithm that links frames with similar latent visual features. After constructing the hypergraph, we proceed to train the Hypergraph Neural Network (HGNN) [21] in parallel with the Structured Attention Layer Mechanism (SALM). This process is mathematically formulated as follows:

$$\widetilde{\mathcal{Y}}_{sri} = \sigma \left( \mathbf{D}_v^{-1/2} \mathbf{H} \mathbf{W} \mathbf{D}_e^{-1} \mathbf{H}^\top \mathbf{D}_v^{-1/2} \cdot \mathbf{F}_{frame\_level} \cdot \Theta \right), \quad (6)$$

where $\widetilde{\mathcal{Y}}_{sri}$ represents the predicted output from the SALM-enhanced HGNN, and $\sigma$ denotes a non-linear activation function, which introduces the necessary non-linearity into the model for capturing complex patterns. $\mathbf{D}_e \in \mathbb{R}^{E \times E}$, $\mathbf{D}_v \in \mathbb{R}^{N \times N}$, and $\mathbf{W} \in \mathbb{R}^{E \times E}$ denote the diagonal degree matrix of hyperedges, the degree matrix of vertices, and weight matrix of hyperedges, respectively. $\mathbf{H} \in \mathbb{R}^{N \times E}$ signifies the incidence matrix that connects hyperedges to their constituent vertices. $\sigma(\cdot)$ denotes the nonlinear activation function (e.g., LeakyReLU($\cdot$)). $\Theta$ is a diagonal matrix representing the learnable parameters updated by the *Cross_Entropy* loss function in the

**Table 3: Results of different models on Subjectivity task (Engagement, Emotion, and EMR Duration). Using the Frame Sequence for Video Representation (FSVR) strategy is denoted by a "△".**

| Models | Protocol | Settings | Engagement (2 classes) | | Emotion (3 classes) | | EMR Duration (3 classes) | |
|---|---|---|---|---|---|---|---|---|
| | | | Acc | F1 | Acc | F1 | Acc | F1 |
| **Random** | P1 | — | 50.44 | 49.93 | 32.30 | 26.26 | 35.01 | 32.10 |
| | P2 | — | 50.14 | 50.00 | 33.13 | 33.03 | 33.52 | 33.18 |
| **GPT4V△** [1] | P1 | Zero-shot | 58.57 | 71.95 | 52.46 | 50.67 | 49.94 | 53.43 |
| | P2 | Zero-shot | 45.62 | 61.53 | 36.40 | 43.65 | 39.39 | 47.04 |
| **Gemini-pro-vision△** [66] | P1 | Zero-shot | 59.89 | 73.70 | 17.66 | 20.00 | 46.40 | 47.96 |
| | P2 | Zero-shot | 46.16 | 63.31 | 30.56 | 43.10 | 36.20 | 43.96 |
| **Video-LLaVA** [47] | P1 | Zero-shot | 60.06 | 74.50 | 61.39 | 71.30 | 45.26 | 57.48 |
| | P2 | Zero-shot | 46.38 | 61.38 | 31.04 | 42.71 | 31.56 | 49.30 |
| **Video-LLaVA** [47] | P1 | Finetune | 66.29 | 66.85 | 72.33 | 81.94 | 61.05 | 61.80 |
| | P2 | Finetune | 52.58 | 52.69 | 38.62 | 44.72 | 41.28 | 50.84 |
| **Video-Chat2** [42] | P1 | Finetune | 75.34 | 76.95 | 71.36 | 75.78 | 57.39 | 60.80 |
| | P2 | Finetune | 60.06 | 60.02 | 39.66 | 40.24 | 44.06 | 45.51 |
| **HMLLM (Ours)** | P1 | Finetune | **78.41** | **79.26** | **78.41** | **84.83** | **62.05** | **62.43** |
| | P2 | Finetune | **64.43** | **64.65** | **43.20** | **48.84** | **51.96** | **56.24** |

**Table 4: Comparative performance of different models on the Objectivity task. Using the FSVR strategy is denoted by a "△". The underline of GPT4V denotes the upper bound. We compute the Accuracy (Acc) and VideoChatGPT-Score (Score) [53] of the proposed method HMLLM and other compared state-of-the-art methods on testing data.**

| Models | Settings | Acc | Score [53] |
|---|---|---|---|
| GPT4V△ | Zero-shot | 84.80 | 3.99 |
| Gemini-pro-vision△ | Zero-shot | 27.15 | 2.35 |
| Video-LLaVA [47] | Zero-shot | 15.20 | 2.06 |
| Video-Chat2 [42] | Zero-shot | 21.80 | 2.11 |
| Video-LLaVA [47] | Finetune | 44.76 | 3.03 |
| Video-Chat2 [42] | Finetune | 49.27 | 3.12 |
| **HMLLM (Ours)** | Finetune | **50.52** | **3.13** |

**Table 5: Results of video conversation [53]. CI: Correctness of Information, DO: Detail Orientation, CU: Contextual Understanding, TU: Temporal Understanding, C: Consistency.**

| Models | CI | DO | CU | TU | C | Avg. |
|---|---|---|---|---|---|---|
| Video LLaMA [81] | 1.96 | 2.18 | 2.16 | 1.82 | 1.79 | 1.98 |
| Video Chat [41] | 2.23 | 2.50 | 2.53 | 1.94 | 2.24 | 2.29 |
| LLaMA Adapter [82] | 2.03 | 2.32 | 2.30 | 1.98 | 2.15 | 2.16 |
| Video-ChatGPT [53] | 2.40 | 2.52 | 2.62 | 1.98 | 2.37 | 2.38 |
| Video-Chat2 [42] | 3.02 | **2.88** | 3.51 | **2.66** | 2.81 | 2.98 |
| **HMLLM (Ours)** | **3.12** | 2.86 | **3.52** | 2.61 | **2.91** | **2.99** |

fine-tuning loop. It functions similarly to a multilayer perceptron (MLP) layer. Finally, $\mathbf{F}_{frame\_level}$ represents the input feature vectors associated with the vertices of the hypergraph. By employing this formulation, we effectively leverage the structural complexity of the hypergraph to enhance the learning capabilities of the HGNN, enabling it to capture and utilize the intricate relationships inherent within the data. This joint training regimen integrates two loss functions: the Cross-Entropy loss ($\mathcal{L}_{CE}$) and the Image-grounded Text Generation (ITG) loss from the prior stage. The combined loss function is expressed as:

$$\mathcal{L} = \mathcal{L}_{ITG} + \lambda \cdot \mathcal{L}_{CE}, \tag{7}$$

where $\lambda$ is a hyperparameter that balances the influence of the Cross-Entropy loss and the ITG loss on the overall optimization process. This composite loss function ensures that the model not only generates text that is grounded in the visual content but also adheres to the learned high-order relationships within the hypergraph structure. This enhances the model's capability to capture intricate interactions and dependencies among video frames.

## 5 Experiment

**Metrics.** In our study, the Subjectivity Task of Video-SME is structured in a multiple-choice question (**MC**) format. To evaluate its

performance, we employ Accuracy (**Acc**) and F1 score as our metrics. For the zero-shot evaluation of the Subjectivity Task, we have devised a unique prompt, with detailed information provided in the appendix. The subsequent task, named Objectivity, involves open-ended (**OE**) text generation. For its evaluation, we introduce evaluation measures [53] based on assessments using GPT-3.5 Turbo.

**Implementation Details.** We employ UMT-L [43] as the visual encoder and Vicuna-7B-v0 [12] as the base model for the SRI-Aware Language Model (SALM). Following the methodology of BLIP2 [40], QFormer is configured with 64 queries. Throughout both the inference and training phases, we adhere to the FSVR strategy detailed in Section 3.1, which involves representing each video with 8 key frames. Further details can be found in the appendix.

### 5.1 Intra Testing

*5.1.1 Intra Testing on Video-SME.* As described in Section 3.4, we have designed two tasks on the collected Video-SME dataset, namely Subjectivity and Objectivity.

**Subjectivity task.** In this task, two protocols are encompassed, *i.e.*, P1, and P2. As shown in Table 3, we present our proposed method HMLLM alongside those from renowned MLLMs such as Gemini-pro-vision, GPT4V, Video-LLaVA, and Video-Chat2. The results, observed from top to bottom, indicate:

1) For zero-shot inference, GPT4V and Video-LLaVA significantly outperformed the random across-the-board. Gemini-pro-vision underperformed the Random baseline in predicting the accuracy of

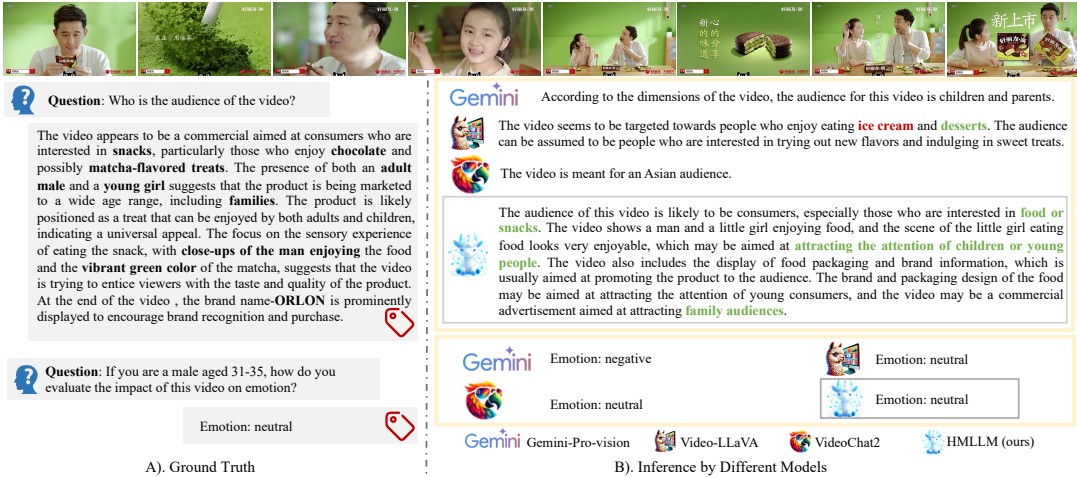

**Figure 4: Qualitative analysis of Video-SME. Green signifies accurate descriptions, while red denotes incorrect responses.**

Engagement and Emotion. Video-Chat2's failure to follow instructions made it difficult to obtain meaningful results. The settings and prompts of zero-shot inference can be found in the appendix.

2) Upon fine-tuning the models with the Video-SME dataset, we observed notable improvements in performance for both Video-LLaVA and Video-Chat2 across both P1 and P2, compared to their initial zero-shot configurations. Moreover, our proposed HMLLM demonstrated the most substantial advancements, consistently outperforming other state-of-the-art methods across all evaluated metrics and protocols.

In detail, within Protocol 1, HMLLM surpassed the leading benchmarks in the categories of Engagement, Emotion, and EMR Duration. The improvements were remarkable, showing enhancements in (accuracy, F1) scores by (3.07, 2.31), (6.08, 2.89), and (1.00, 0.63), respectively. These results underscored the efficacy of our method in accurately capturing and analyzing both engagement and emotional dynamics, as well as predicting EMR duration with high precision. For Protocol 2, the superiority of HMLLM is equally evident. Again, it outshoned the best-existing benchmarks with enhancements in (accuracy, F1) scores by (4.37, 1.34), (3.54, 4.12), and (7.90, 5.40), respectively. These findings highlight the robustness and adaptability of our model across different protocols, establishing its potential for widespread applicability in real-world scenarios.

**Objectivity Task.** In the exploration of the objectivity task, as detailed in Section 3.3, we meticulously refined the ground truth (GT) by manually correcting annotations initially provided by GPT4V. This meticulous process contributed to the notably high zero-shot inference capabilities observed for GPT4V. Given that Gemini-provision and GPT4V inherently lack support for video inputs, we integrated FSVR to bridge this gap. This adaptation endowed both models with the ability to process video inputs, thus expanding their applicability across a wider range of tasks. As shown in Table 4, GTP4V became the upper bound in a zero-shot setting because we semi-automatically utilized it for labeling, as described in Section 3.3. When the narrative shifts upon the fine-tuning of our models with the Video-SME dataset. Both Video-LLaVA and Video-Chat2 showcased enhancements in their performance metrics, surpassing

their initial zero-shot configurations. This improvement highlights the transformative impact of targeted training on model efficacy. Notably, our proposed HMLLM method emerged as a formidable contender, eclipsing other models in performance across the board. Specifically, HMLLM outperformed the best baseline, Video-Chat2, in terms of Acc and the Score [53] by 1.25 and 0.01, respectively.

The results not only validate the effectiveness of fine-tuning with the Video-SME dataset but also emphasize that our HMLLM method sets a new benchmark in model performance.

*5.1.2   Intra Testing on Video Conversation Benchmark.* To further validate the performance of HMLLM, we conducted experiments on other video-based generative performance benchmarks. Following the setup of Video-ChatGPT[53], we present the performance of our proposed HMLLM, detailed in the last row of Table 5. Experimental results demonstrate that the HMLLM effectively enhances both Contextual Understanding and Consistency. Given the HMLLM did not overemphasize temporal details, a slight decrease in Temporal Understanding was observed.

## 5.2   Analysis and Visualization

We further present a qualitative comparison in Figure 4. HMLLM demonstrates an enhanced ability to generate longer and more comprehensive responses for Objectivity Tasks. This improvement can be attributed to the longer average context length of our dataset, which facilitates a deeper understanding of video content by enabling detailed analysis of advertising plots and visual elements. More detailed qualitative analyses are available in the appendix.

## 6   Conclusion

In this paper, we released a large-scale Video-SME dataset with two challenging tasks. We hope it will push cutting-edge research in video understanding. Besides, we proposed a novel HMLLM approach that enhances the language model by constructing a hypergraph feature space across modalities, thereby providing semantically richer associative features. Finally, we conducted a comprehensive set of experiments on both Video-SME and other video-based generative datasets, verifying the significance of the proposed dataset and method.

# Acknowledgments

This work was supported by the Brain-like General Vision Model and Applications project (Grant No. 2022ZD0160403), China Post-doctoral Science Foundation (2023M740079, GZC20230058).

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
