# OpenReview forum: "Hypergraph Multi-modal Large Language Model: Exploiting EEG and Eye-tracking Modalities to Evaluate Heterogeneous Responses for Video Understanding"
_acmmm.org/ACMMM/2024/Conference — MM2024 Oral_

### Official Review · Reviewer_kctG · 2024-05-08

**Rating:** 5
**Confidence:** 3

**Summary:**

The paper proposes a novel large-scale dataset, SRI-ADV, and a Hypergraph Multi-modal Large Language Model (HMLLM) that utilizes EEG and eye-tracking data to enhance video understanding. This work is a substantial contribution to cognitive neuroscience and video content analysis, highlighting the importance of multi-modal data for understanding user engagement and emotional responses.

**Strengths:**

1. The introduction of the SRI-ADV dataset, which integrates EEG and eye-tracking data with video content analysis, addresses a gap in current research. This dataset not only enhances the complexity of the analysis but also provides a comprehensive approach to understanding subjective user engagement.
2. The development of HMLLM, which leverages hypergraph techniques to analyze the relationships among video elements, EEG signals, and eye-tracking data, is interesting. The model's ability to bridge semantic gaps across modalities shows potential for deep insights into user cognition.

**Limitations:**

1. The paper lacks a detailed discussion on the generalizability of the proposed model to other forms of media content beyond advertisement videos. It's unclear how the model performs in different contexts or with different types of video content, which could limit its application in broader scenarios.
2. The collection of data primarily from viewers in Mainland China might introduce cultural biases in model training and interpretation. Expanding the dataset to include a more diverse global audience could enhance the model's robustness and applicability.
3. Integrating additional modalities, such as acoustic signals or textual annotations, could provide a richer understanding of user interactions with video content.

**Suitability:**

2

---

### Official Review · Reviewer_H4Cb · 2024-05-21

**Rating:** 5
**Confidence:** 3

**Summary:**

This paper presents a large-scale video dataset, which captures real-time EEG and eye-tracking data, including 498 videos and 178k Q&A pairs. The authors also introduce a multi-modal large language model that leverages EEG and eye-tracking data to assess heterogeneous responses to video content. Extensive experiments on the SRI-ADV dataset and additional benchmarks demonstrate the model's effectiveness.

**Strengths:**

1.	The proposed large-scale dataset will be useful and benefit the community.
2.	Extensive experiments demonstrate the effectiveness of proposed method.
3.	Ablation studies verify that each component is an optimal solution.

**Limitations:**

1.	The subjective response indicator is pivotal in this paper; however, there is a need for a clearer definition of what it specifically encompasses. It is unclear whether it includes emotions, engagement levels, or other relevant factors such as user satisfaction, attention span, or cognitive load. Providing a more detailed explanation of the components of the subjective response indicator would enhance the reader's understanding and the overall impact of the study.
2.	Video emotion analysis is a well-studied area, and the proposed method actually involves emotion in dataset construction and methodology design. However, there is no discussion or comparison involving video emotion analysis. This may lead the audience to misunderstand the contribution of this work.

[1] Representation Learning through Multimodal Attention and Time-Sync Comments for Affective Video Content Analysis, ACM MM22

[2] Weakly Supervised Video Emotion Detection and Prediction via Cross-Modal Temporal Erasing Network, CVPR23

[3] Towards Robust Multimodal Sentiment Analysis under Uncertain Signal Missing, SPL23

**Suitability:**

3

---

### Official Review · Reviewer_7FwQ · 2024-05-24

**Rating:** 5
**Confidence:** 3

**Summary:**

The paper presents a comprehensive study on a Hypergraph Multi-modal Large Language Model (HMLLM), developed to enhance video understanding by integrating multiple data modalities, including EEG signals and eye-tracking data. The paper provides these main components:
1.	SRI-ADV Dataset: Introduction of a large-scale dataset, that captures EEG and eye-tracking data from diverse demographics while watching advertisement videos.
2.	HMLLM Algorithm: Hypergraph Multi-modal Large Language Model that employs hypergraph structures to encode complex relationships among video elements, EEG signals, and eye-tracking data.
3.	Results: Extensive experimental evaluations demonstrating the effectiveness of the proposed model on the SRI-ADV dataset and additional video-based generative datasets.

**Strengths:**

1.	Innovative Dataset: The SRI-ADV dataset fills a gap in video understanding research by providing rich modality information and comprehensive Q&A pairs, allowing for the assessment of video creativity and implicit factors.
2.	Model Architecture: The use of hypergraphs to represent complex relationships among different data modalities is fits well in this problem as its ability to encapsulate logical reasoning and semantic analysis based on different multimodalities.
3.	Comprehensive Analysis: The paper includes thorough experimental evaluations and qualitative comparisons, that demonstrate the model's ability to generate longer, more comprehensive responses for objectivity tasks.
4.	Writing of the paper. The paper is well-written and easy to follow.

**Limitations:**

-	Heterogeneous Data Types: Despite the importance of mixing these modalities for understanding subjects’ responses while watching ads videos, EEG signals, eye-tracking data, and video content are fundamentally different types of data, each with unique characteristics and formats. Integrating these heterogeneous data types into a unified representation is inherently complex. It is  unclear in the paper how the authors achieved the temporal and spatial alignment between these data streams to accurately capture the relationships between them requires sophisticated preprocessing and alignment techniques. Misalignment can lead to incorrect or misleading interpretations of the data.

-	Computational power of large scale data: Constructing hypergraphs for large scale datasets, like the SRI-ADV dataset, requires substantial computational power and memory. I think the authors need to shed some light on this aspect in the paper.

**Suitability:**

3

---

### Official Review · Reviewer_XfSk · 2024-05-25

**Rating:** 5
**Confidence:** 2

**Summary:**

The paper introduces the Hypergraph Multi-modal Large Language Model (HMLLM) designed to integrate and analyze multi-modal data, including Electroencephalographic (EEG) and eye-tracking signals, for understanding video content. The authors present a novel large-scale dataset, Subjective Response Indicators for Advertisement Videos (SRI-ADV), which captures real-time EEG and eye-tracking data from diverse demographics watching advertisement videos. This dataset aims to bridge the gap in video understanding by providing rich modality information and a comprehensive set of question-and-answer (Q&A) pairs. The proposed HMLLM utilizes hypergraph learning to represent complex relationships among video elements, EEG signals, and eye-tracking data, enhancing logical reasoning and semantic analysis.

**Strengths:**

The introduction of the SRI-ADV dataset is a significant contribution, offering a rich, multi-modal dataset that includes real-time EEG and eye-tracking data. This dataset addresses the limitations of existing video understanding benchmarks by providing more modalities.

The use of hypergraph learning to represent and analyze complex relationships among different modalities is a novel approach that enhances the model's ability to perform logical reasoning and semantic analysis.

The experimental evaluations demonstrate the effectiveness of HMLLM, showing improvements over existing models.

**Limitations:**

The paper does not explicitly mention the computational efficiency or complexity analysis of the HMLLM model. However, given the integration of multiple modalities and the use of hypergraph learning, it is reasonable to infer that the model might be computationally intensive, potentially requiring significant resources for training and fine-tuning.

While the model performs well on the SRI-ADV dataset, its generalization to other types of video content and different demographic data is only briefly discussed in Section 5.1.2. Further experiments on varied datasets would strengthen the validation of the model.

**Suitability:**

3

---

### Meta-Review · Area_Chair_Tb2E · 2024-07-01

**Recommendation:** Accept (Oral)
**Confidence:** 4

**Metareview:**

The initial reviews were positive with 4x weak accept. After the rebuttal, one reviewer upgraded their review to a strong accept, while one other confirmed their weak accept.

The reviewers agree in this submission including (+) a innovative dataset and (+) having novelty in the use of hypergraph learning.

While there are some minor concerns on (-) the generalizability of subjective data, including cultural biases through the dataset being Chinese, the overall concerns seem limited hence the positive ratings.

I believe this paper should definitive be accepted.